# Vehicle-Free Nanotheranostic Self-Assembled from Clinically Approved Dyes for Cancer Fluorescence Imaging and Photothermal/Photodynamic Combinational Therapy

**DOI:** 10.3390/pharmaceutics14051074

**Published:** 2022-05-17

**Authors:** Mingbin Huang, Chao Xu, Sen Yang, Ziqian Zhang, Zuwu Wei, Ming Wu, Fangqin Xue

**Affiliations:** 1Shengli Clinical Medical College, Fujian Medical University, Fuzhou 350001, China; hmb6140103161@fjmu.edu.cn (M.H.); xc9201851054@fjmu.edu.cn (C.X.); zzq6150103134@fjmu.edu.cn (Z.Z.); 2Department of Gastrointestinal Surgery, Fujian Provincial Hospital, Fuzhou 350001, China; 3The United Innovation of Mengchao Hepatobiliary Technology Key Laboratory of Fujian Province, Mengchao Hepatobiliary Hospital of Fujian Medical University, Fuzhou 350025, China; N190820053@fzu.edu.cn

**Keywords:** vehicle-free nanotheranostic, fluorescence imaging, photothermal/photodynamic combinational therapy, indocyanine green, methylene blue

## Abstract

Phototherapy, including photothermal therapy (PTT) and photodynamic therapy (PDT) has attracted growing attention as a noninvasive option for cancer treatment. At present, researchers have developed various “all-in-one” nanoplatforms for cancer imaging and PTT/PDT combinational therapy. However, the complex structure, tedious preparation procedures, overuse of extra carriers and severe side effects hinder their biomedical applications. In this work, we reported a nanoplatform (designated as ICG-MB) self-assembly from two different FDA-approved dyes of indocyanine green (ICG) and methylene blue (MB) without any additional excipients for cancer fluorescence imaging and combinational PTT/PDT. ICG-MB was found to exhibit good dispersion in the aqueous phase and improve the photostability and cellular uptake of free ICG and MB, thus exhibiting enhanced photothermal conversion and singlet oxygen (^1^O_2_) generation abilities to robustly ablate cancer cells under 808 nm and 670 nm laser irradiation. After intravenous injection, ICG-MB effectively accumulated at tumor sites with a near-infrared (NIR) fluorescence signal, which helped to delineate the targeted area for NIR laser-triggered phototoxicity. As a consequence, ICG-MB displayed a combinational PTT/PDT effect to potently inhibit tumor growth without causing any system toxicities in vivo. In conclusion, this minimalist, effective and biocompatible nanotheranostic would provide a promising candidate for cancer phototherapy based on current available dyes in clinic.

## 1. Introduction

Tumors are one of the major public health problems that seriously threaten human health and social development [1]. In recent years, the incidence of cancer has shown a high growth trend worldwide. In 2020, it was projected that there would be about 19.3 million new cancer patients worldwide and nearly 10 million deaths [2]. At present, the treatment of tumors is still mainly based on surgery, radiotherapy and chemotherapy [3,4]. However, the vast majority of patients with advanced tumors are not suitable for surgical resection, and some cancer cells may remain or recur after resection. Meanwhile, chemotherapy and radiotherapy are often associated with highly resistant or insensitive cancer cells, as well as the inevitable damage to the normal tissues and cells [5,6,7]. Therefore, it is necessary to develop new tumor treatment methods with remarkable curative effect and minimal side effects.

Phototherapy, including photothermal therapy (PTT) and photodynamic therapy (PDT), is a new tumor treatment paradigm that has attracted growing interest over the past decades [8,9]. PTT uses external near-infrared light to irradiate the tumor site, so that the exogenous photothermal conversion agents (PTCAs) in the tumor irradiation area absorb the energy of photons and convert it into heat energy, thereby producing hyperthermia to kill tumor cells [9,10,11,12]. In contrast, PDT generates highly cytotoxic reactive oxygen species (ROS) like ^1^O_2_ to destroy cancer cells by utilizing the photochemical reactions of photosensitizers (PSs) [13,14]. Different from the traditional treatment strategies, phototherapy has the advantages of minimal invasiveness, spatiotemporal controllability and negligible drug resistance. Many phototherapeutic agents, especially PSs, with a tetrapyrrole structure have been used for PDT in clinic [15]. Nevertheless, its therapeutic effect is severely hindered by laser attenuation, the limited bioavailability of PSs and the hypoxic tumor environment [16]. Different from PDT, PTT is oxygen-independent but also has the problem of incomplete tumor ablation due to inhomogeneous heat distribution and protective effect of heat shock proteins (HSPs) [17]. Since the advantages of PDT and PTT can be integrated and complemented with each other, their combinational manner might improve the therapeutic efficacy at low dose or low laser density, thus reducing the side effects to healthy tissues.

The development of nanotechnology provides a good choice for simultaneously delivering PTCAs and PSs through “all-in-one” nanocarriers to realize combinational PTT/PDT and other functions. At present, researchers have developed many nanosystems for tumor PTT/PDT, such as gold nanoparticles, bismuth ferrite nanoparticles, metal-organic frameworks, etc., [18,19,20]. These nanosystems have achieved great progress in antitumor therapy, but there are still some problems that need to resolve urgently, such as poor degradability or toxic metabolites after degradation, complicated design, tedious synthetic procedures, and complex structure. Worse yet, the limited drug encapsulating capacity would cause the overuse of excipients to maintain a therapeutic dose, thus raising the potential system risk [21]. One approach to overcome these limitations is to develop self-delivery nanosystems (SDNSs) constructed from pure active drugs without the assistance of any additional carriers [22,23,24]. However, in these designs, how to seek and align therapeutic agents with different structures and physicochemical features into “all-in-one” SDNSs for the sake of exerting certain collaborative functions remains an unmet challenge, especially considering the future application in clinic. As far as we know, SDNSs self-assembled from clinically used agents for combinational PTT/PDT have been rarely reported. 

ICG and MB are two typical fluorophores approved by FDA for NIR fluorescence imaging of lymph node, tumor, tissue perfusion and other vital structures [25]. In addition, they also exhibit good photothermal conversion [26,27,28,29] and photodynamic effects for PTT and PDT, respectively [30,31,32]. Taking advantage of the passive accumulation in tumor tissues by the enhanced permeability and retention (EPR) effect, we and others developed ICG or MB-based nanosystem to improve their phototheranostic performance by increasing photostability, enhancing target specificity, and prolonging circulation lifetime [33,34,35]. To further minimize the nanosystem, we herein reported a carrier-free nanoplatform of ICG-MB self-assembled from ICG and MB for NIR fluorescence imaging and the combinational PTT/PDT of colon cancer (Figure 1). The formation, constitution, dispersion and other physico-chemical properties of ICG-MB were characterized by scanning electron microscopy (SEM), dynamic light scattering (DLS) analysis, FT-IR and UV-vis-NIR spectroscopy, etc. The photothermal conversion and ^1^O_2_ generation abilities of ICG-MB nanoparticles were investigated and compared with free ICG and free MB in aqueous solution to verify their superiority in terms of improving photostability as an entirety under 808 nm and 670 nm laser irradiation. Afterwards, the potential of ICG-MB as a nanothernostic agent for NIR fluorescence imaging and PTT/PDT was evaluated both in vitro and in vivo by using colorectal cancer cells (CT26) as the model. Finally, the systemic toxicity of ICG-MB in vivo was checked by hematological and histological analysis.

## 2. Materials and Methods

### 2.1. Materials

Methylene blue was purchased from J&K Scientific Co. Ltd. (Beijing, China). Indocyanine green was obtained from Tokyo Chemistry Industry Co. Ltd. (Tokyo, Japan). Dimethyl sulfoxide was obtained from Sigma-Aldrich Inc. (Saint Louis, MO, USA). An Annexin V-FITC/PI apoptosis detection kit and Cell Counting Kit-8 (CCK-8) were purchased from Dojindo Laboratories (Kumamoto, Japan). A Singlet oxygen sensor green reagent (SOSG) was purchased from Molecular Probes Inc. (Eugene, OR, USA). Ultrapure water was manufactured by a Milli-Q Gradient System (18.2 M Ω resistivity, Millipore Corporation, Bedford, MA, USA). If not specified, all other chemicals were commercially available and used as received.

### 2.2. Cell Culture

CT26 cells (colorectal cancer cells) and NCM365 cells (colon epithelial cells) were purchased from ATCC (Manassas, VA, USA). Cells were cultured in DMEM which contained 10% fetal bovine serum, 1% penicillin-streptomycin in a 37 °C incubator equipped with a humidifier and 5% CO_2_.

### 2.3. Synthesis of ICG-MB NPs

ICG-MB nanoparticles were prepared by nanoprecipitation methods. In brief, 20 mg ICG and 10 mg MB dissolved in 2.5 mL DMSO, then added into 10 mL ultrapure water dropwise under gentle stirring. After 4 h of constant stirring, the mixed solution was first centrifuged at 3000× *g* for 10 min to remove big aggregates in precipitate. The residual supernatant was further centrifuged at 20,000× *g* for 30 min to obtain ICG-MB crude product, followed by washing twice with ultrapure water. At last, the as-prepared product was dispersed in water for further usage.

### 2.4. Characterization of ICG-MB NPs

The morphologies of nanoparticles were observed using scanning electron microscopy (SEM, JEOL, Japan). The UV-vis-NIR absorbance and fluorescence spectrum were measured by a Spectra Max M5 microplate reader (Molecular Devices, San Jose, CA, USA) and a Cary Eclipse fluorescence spectrophotometer (Agilent Technologies, Selangor Darul Ehsan, Malaysia), respectively. The 670 nm and 808 nm laser (Diode Laser System, Changchun Laser Optoelectronics Technology Co., Ltd., Changchun, Jilin, China) was used to induce photodynamic therapy (PDT) and photothermal therapy (PTT), respectively. The images of cell uptake were obtained by a CLSM (Zeiss LSM780, Shanghai, China). LIVE/DEAD assay and ROS generation assay were observed by a fluorescence microscopy (Zeiss Axio Vert.A1, Baden-Württemberg, Germany). The mice fluorescence imaging was performed on the UniNano-NIR II system.

### 2.5. Evaluation of the Photothermal Ability of ICG-MB NPs

In order to examine the photothermal activity, ICG-MB was prepared at varying concentrations (0, 10, 20, 40, 60 μg/mL) and then treated with laser irradiation (808 nm, 1 W/cm^2^) for 1000 s. Furthermore, for the evaluation of the photothermal stability, the temperature changes were monitored at 1000-s interval (1000-s laser on and 1000-s laser off) for four irradiation cycles (1 W/cm^2^, 808 nm). The IR thermal camera (Ti25 Fluke Co., Everett, WA, USA), along with the thermocouple microprobe (Φφ = 0.5 mm) (STPC-510P, Physitemp Instruments LLC., Shanghai, China) worked on monitoring and recording the fluctuation of temperature. The photothermal conversion efficiency (η) was also calculated according to literature reports: (1)$$\;\eta \;=\frac{\left(hA\Delta Tmax-Qs\right)}{I(1-10^{-A_{\lambda })}}=\frac{\left(4.2/287.92\right)\times 34.9-0.0252}{0.8\times \left(1-10-0.9465\right)}\times 100\%=68.2\%$$

### 2.6. Photodynamic Performance of ICG-MB NPs

Since it has been well proved that the photodynamic ability of MB is of high quality, we ran the following test to investigate whether ICG-MB could inherit it from its raw material, 1 mL ICG-MB (10 μg/mL) containing 10 μΜ Singlet Oxygen Sensor Green Reagent (SOSG) was exposed to the laser radiation (670 nm) for different times (0, 2, 4, 6, 8, 10 min), and the fluorescence spectrum from 500 to 750 nm was recorded under 480 nm wavelength excitation. The ^1^O_2_ generation efficiency was further analyzed by calculating F_t_/F_0_ at 530 nm, where F_0_ refers to the initial fluorescence intensity, while F_t_ refers to the fluorescence intensity after irradiation at different times. 

### 2.7. Cellular Internalization

CT26 cells were seeded in a confocal dish at a density of 2 × 10^4^ cells per dish and incubated in darkness for 24 h. Then the cells were incubated with PBS, ICG-MB or the physical mixture of ICG and MB (ICG&MB), in which the ICG concentration was set at 40 μg/mL. For another 24 h, PBS was used to wash cells twice. Afterward, the cells were washed with PBS, fixed with 4% paraformaldehyde, stained with DAPI, and finally observed under CLSM (Zeiss LSM780).

### 2.8. Intracellular ROS Detection

To verify that the laser could stimulate cells to release reactive oxygen species once CT26 cells uptake ICG MB, we set up the following experiment. Cells were seeded in a 96-well plate at a density of 6 × 10^3^ cells per well and incubated away from light for 24 h, then medium containing either ICG-MB or PBS was substituted for the original medium, both of whose MB concentrations are equally 1 μg/mL. After 24 h of incubation, the cells were washed with PBS and the medium containing 40 µM DCFH-DA (an intracellular ROS detection probe) was added for another 15 min of incubation. Next, the cells were washed twice with PBS and irradiated under a 670 nm laser (20 mW/cm^2^) for 3 min. Finally, the cells were observed under a fluorescence microscope (Zeiss Axio Vert. A1, Baden-Württemberg, Germany).

### 2.9. In Vitro Cytotoxicity Assay 

The cytotoxicity of ICG-MB NPs was evaluated in cancer cell lines (CT26 cells) and normal cell lines (NCM356 cells) by CCK-8 assay. After being seeded in a 96-well plate at a density of 1.2 × 10^4^ cells per well and incubated for 24 h, cells were treated with varying concentrations of ICG-MB (0, 10, 20, 30, 40 μg/mL). The cells were washed twice with PBS 24 h later. Meanwhile, untreated cells were used as a control group. Furthermore, 100 μL fresh medium with 10% CCK-8 solution was added. 90 min later, we measured the absorbance at 450 nm of each well with a microplate reader (Molecular Devices, San Jose, CA, USA). The viability was calculated as literature mentions: Cell viability (%) = (A_1_ − A_blank_)/(A_0_ − A_blank_) × 100%

### 2.10. In Vitro Cell Killing Ability

CT26 cells were seeded in a 96-well plate at a density of 1.2 × 10^4^ cells per well and incubated for 24 h, and then the original medium was replaced with a fresh one containing different concentrations of ICG-MB (0, 10, 20, 30, 40, μg/mL). Twenty-four hours later, the cells were exposed to different laser radiations (670 nm 20 mW/cm^2^, 808 nm 1 W/cm^2^) for 5 min. After another 24 h incubation, the CCK-8 assay kit was used to measure the cell viability.

To further confirm phototoxicity, a LIVE/DEAD Cells assay was also carried out. Briefly, the cells with the aforementioned treatment were stained with Calcium AM and PI (2 μL AM and 2 μL PI in every 1 mL medium) for 15 min and imaged by a fluorescence microscope.

Moreover, the PTT/PDT anticancer effect was also evaluated by Annexin V-FITC/PI apoptosis assay. The treated cells were stained with Annexin V-FITC/PI solution and detected by flow cytometry.

### 2.11. Animal Experiments 

All animal experiments were performed in accordance with the guidelines approved by the Animal Ethics Committee of Mengchao Hepatobiliary Hospital of Fujian Medical University and were conducted according to the institutional guidelines. The BALB/c nude mice were purchased from China Wushi, Inc. (Shanghai, China). A CT26 subcutaneous tumor model was constructed by subcutaneous injection of CT26 cells (1 × 10^6^) in the rear flanks of mice. When the tumor grew to approximately 100–150 mm^3^, the tumor bearing mice were prepared for in vivo biodistribution and antitumor therapeutic experiments.

### 2.12. Biodistribution of ICG-MB NPs

To explore the biodistribution in vivo, CT26 tumor-bearing mice were intravenously (i.v.) injected with ICG-MB at a dose of 100 μL (5 mg/mL) per mouse. A UniNano- NIR-II system was then used to monitor the fluorescent signals of mice at 4, 6, 8, 10, 24, and 48 h post injection. After 48 h of injection, the mice were sacrificed by CO_2_ inhalation and the main organs were resected and imaged.

### 2.13. In Vivo Anti-tumor Ability of ICG-MB NPs

To directly reveal the anti-tumor performance, CT26 tumor-bearing mice were randomly divided into 5 groups (*n* = 5, each group) and i.v. injected with PBS or ICG-MB (4 μg/mL), then treated with different laser radiations (670 nm 20 mW/cm^2^, 808 nm 1 W/cm^2^), and thermal images were collected. The tumor volumes and mice body weights were measured every other day. The tumor volumes were calculated as length × width^2^/2. At day 20, the tumors and major organs were excised from the mice. The tumors were weighed and recorded after being fixed in formalin for H&E and Ki67 staining. We also evaluated the toxicity of ICG-MB by serum biochemistry analyses.

### 2.14. Statistical Analysis

Data was presented as the mean ± standard deviation (SD) or min to max, showing all points as indicated. Significance was calculated using a one-way analysis of variance (ANOVA) or *t* test as indicated. Prism 6 software (GraphPad) was used to perform all statistical analyses. * *p* < 0.05, ** *p* < 0.01, *** *p* < 0.001. A *p*-value < 0.05 was considered as statistically significant.

## 3. Results and Discussion

### 3.1. Synthesis and Characterization of ICG-MB NPs 

In this work, ICG-MB nanosystems were synthesized via a facile nano-precipitation method at the optimized feed ratio of 2:1 (ICG:MB, W/W) to provide a stable nanoparticulated dispersion with a distinct Tyndall phenomenon and without aggregation after storage over 24 h. As depicted in Figure 1a, ICG and MB are predicted to self-assemble into an integrated entity by π-π stacking, electrostatic and hydrophobic interactions [23]. The morphology was analyzed by scanning electronic microscopy (SEM). From the SEM images, it is clearly seen that the obtained ICG-MB exhibited an irregular aggregate structure of primary spherical nanoparticles and good monodispersity, with a diameter around 100 nm (Figure 1b). For reflecting the actual dimension of ICG-MB in the aqueous solution, the hydrodynamic diameter of ICG-MB nanoparticles was tested through dynamic light scattering (DLS) measurement and determined to be 142.3 ± 0.7 nm (Figure 1c), which was a suitable size for preferential tumor accumulation through the EPR effect. Meanwhile, the surface potential of ICG-MB peaked around the neutral range (Figure 1d), which might have resulted from the electrostatic charge neutralization of positively charged MB and negatively charged ICG. From UV-vis-NIR spectra in Figure 1e, ICG-MB had a widened and red shifted band in comparison with free ICG and free MB, indicating the supramolecular interactions such as hydrophobic and π-π stacking interactions between ICG and MB in ICG-MB nanoparticles, which is consistent with the molecular docking analysis in Figure 1a [36]. To further confirm the chemical component of ICG-MB, FT-IR spectra were recorded in Figure 1f. The characteristic peaks at 1353 cm^−1^ (sulfonic group) and 1420 cm^−1^ (C=C stretches in long chain alkenes) in ICG [37], as well as the peak at 1591 cm^−1^ (C=N stretches in central ring) in MB [38], all appeared in the spectrum of ICG-MB. These results demonstrated the successful formation of hybrid nanomedicine self-assembly from ICG and MB.

Additionally, the hydrodynamic size of ICG-MB was not found to vary dramatically when stored in PBS with 10% of FBS for six days, without any obvious aggregations/precipitations (Figure 1g). Meanwhile, the absorption spectra of ICG-MB also did not change after being stored for six days, further verifying the stability of this nanoagent (Figure 1h). This result verified a favorable stability in the physiological environment for the following applications. To further determine the content of ICG and MB in ICG-MB, the lyophilized product was dissolved in DMSO for UV-vis measurement at 800 nm. From the standard curve of ICG, it is calculated that there was 57.1% of ICG in ICG-MB, thus the MB content was estimated to be 42.9%.

### 3.2. Photothermal Properties of ICG-MB

Since ICG is one of the most frequently used PTCAs, the photothermal effect of ICG-MB was evaluated in the NIR region (Figure 2a). To investigate the photothermal properties in more detail, the aqueous solution of ICG-MB with different concentrations (0 μg/mL, 10 μg/mL, 20 μg/mL, 40 μg/mL, and 60 μg/mL) was exposed to an 808 nm NIR laser (1 W/cm^2^). As shown in Figure 2b, the concentration and time-dependent temperature raising effects of ICG-MB were detected with the laser irradiation. The photothermal conversion efficiency (η) of ICG-MB was calculated to be 68.2% (Figure 2c,d), which was higher than most of the previously reported PTT agents, such as Au nanorods (21%) [39], Cu_2-x_Se nanocrystals (22%) [40], Bi_2_S_3_ nanorods (28.1%) [41], etc. To further identify which component is contributing to the photothermal effect, we compared the temperature elevation of ICG-MB (containing 34.3 μg/mL of ICG and 25.7 μg/mL of MB), free ICG (34.3 μg/mL), free MB (25.7 μg/mL) and pure water during an 808 nm laser irradiation (1 W/cm^2^). As displayed in Figure 2e, the temperature of ICG-MB and free ICG showed an obvious increase trend in comparison with free MB and pure water without significant variation under laser irradiation, which indicated that the photothermal conversion ability of ICG-MB resulted from ICG rather than MB. Importantly, ICG-MB was more pronounced in terms of NIR laser-induced heat conversion than free ICG. In addition, compared with free ICG that nearly lost all photothermal conversion capacity, ICG-MB still kept a good temperature increase performance after four laser on/off cycles, highlighting the potential of ICG-MB NPs as a durable photothermal agent for PTT cancer treatment (Figure 2f). The excellent photothermal stability could be explained by the high resistance to photobleaching of ICG-MB nanoparticles upon 808 nm laser irradiation via an activated quenched state possibly due to intermolecular dye interactions [42], whereas the absorbance of free ICG declined dramatically with the same treatment (Figure 2g–i). Together, these results clearly suggest the overwhelming superiority of ICG-MB to generate heat in PTT treatment.

### 3.3. Photodynamic Properties of ICG-MB 

The wide application of MB in PDT motivated us to explore the ^1^O_2_ generation after encapsulation into ICG-MB under 670 nm laser irradiation. Toward this end, the singlet oxygen sensor green reagent (SOSG) was used as a chemical trapping probe whose fluorescence signal would irreversibly increase once in the presence of ^1^O_2_ [43]. As displayed in Figure 3a,b, the fluorescence intensities increased rapidly after 670 nm laser irradiation in free MB and ICG-MB with a time-dependent manner. Although ICG-MB showed a lower ^1^O_2_ generation rate rather than free MB due to the aggregation quenching effect [44], this nanotheranostic nanoparticle maintained a nearly linear time-dependence of ^1^O_2_ generation after 10 min of continuous irradiation while no further ^1^O_2_ generation was found in free MB after 4 min of irradiation (Figure 3c). To explain this phenomenon, we further compared the absorbance changes of free MB and ICG-MB upon laser irradiation. As depicted in Figure 3d–f, ICG-MB exhibited a good photostability with negligible absorption decline for 30 min of continuous laser irradiation. In contrast, the same laser irradiation severely impaired the absorption ability of free MB. Thus, in addition to ICG, ICG-MB nanoparticles could also improve the photostability of free MB to continuously or repeatedly produce ^1^O_2_ for PDT treatment.

### 3.4. Intracellular Internalization and ROS Production

To study the cellular uptake of the nanoplatform, CT26 cells (colorectal cancer, the third most common malignancy [45]) were incubated with ICG-MB or the physical mixture of free ICG and free MB (ICG&MB) at the same concentration. After 4 h of incubation, the cells were imaged by CLSM. As shown in Figure 4a, the fluorescence signal of ICG in the cytoplasm region of cells treated with ICG&MB was not obvious compared to that in ICG-MB treated cells, which revealed that the assembly of ICG and MB into the nanosystem was able to effectively promote intracellular ICG transport. This phenomenon might be ascribed to the strengthened interactions between cell membranes and the ICG-MB bearing a near-neutral surface, while the free ICG do not easily contact the cell membrane due to their negatively charged characters [46].

Efficient ROS production in cancer cells is vital to realize PDT. Next, the intracellular production of ^1^O_2_ photoinduced by ICG-MB in colorectal cancer cells of CT26 was assessed using 2′,7′-dichlorofluorescin diacetate (DCFH-DA) as an indicator. Initially, DCFH-DA is nonfluorescent. However, ROS can fleetly oxidize it to form 2′,7′-dichlorofluorescein (DCF) with highly green fluorescence. As shown in Figure 4b, the strong DCF fluorescence signal was observed in CT26 cells treated with ICG-MB following irradiation with a 670 nm laser. In contrast, the cells only treated with ICG-MB or 670 nm laser irradiation showed negligible ^1^O_2_ generation (Figure 3b). This data demonstrated that ICG-MB could generate ROS abundantly for efficient PDT under the irradiation of a 670 nm laser.

### 3.5. In Vitro Cell Cytotoxicity and Phototherapy Effect

The safety of nanoparticles was first investigated by CCK-8 assay. ICG-MB nanoparticles at different equivalent ICG concentrations (0, 10, 20, 30, 40 μg/mL) were incubated with colonic epithelial cells of NCM356 for 24 h. The cell viability was above 85% at all tested concentrations, which showed a low toxicity against normal cells (Figure 5a). Since NIR-mediated ICG-MB has both photothermal conversion and ROS generation ability as an integrated entity, we next investigated these phototherapeutic effects toward CT26 cells in vitro. The CT26 cells incubated with different equivalent ICG concentrations of ICG-MB (0, 1, 2, 30, 40 μg/mL) and irradiated with different lasers for 5 min: (1) no laser, (2) 670 nm, (3) 808 nm, (4) 670 + 808 nm. As shown in Figure 5b, only laser irradiation without ICG-MB (0 μg/mL) or only ICG-MB treatment without laser irradiation had a limited phototherapy effect on CT26 cell. Once exposed to 670 nm or 808 nm light, ICG-MB showed a dose-related phototoxicity due to the PDT and PTT effect, with the cell viability decreasing to 29.8% and 24.4% at the concentration of 40 μg/mL, respectively. Most significantly, the incorporation of a 670 nm and 808 nm laser could further reinforce the anti-cancer effect of ICG-MB, due to the combinational effect of PDT and PTT. The therapy effect of ICG-MB was also visually confirmed by LIVE/DEAD cell staining. Tumor cells were divided into five groups and underwent following treatments: (1) PBS, (2) ICG-MB, (3) ICG-MB + 670 nm, (4) ICG-MB + 808 nm, and (5) ICG-MB + 670 nm + 808 nm. As shown in Figure 5c, it was found that the number of live cells (green) was overwhelming in groups (1) and (2), which again proved that our as-prepared ICG-MB nanoparticles had good biosafety. As expected, single PDT (group 3) or single PTT (group 4) showed a moderate cell-killing effect. However, almost all of the cells were found dead with an entirely red fluorescence signal in group 5 with ICG-MB incubation in the presence of both 670 nm and 808 nm light. These results certified the feasibility of ICG-MB as a PDT/PTT agent for cancer cell ablation. Finally, a flow cytometry assay was then employed to analyze the cell apoptosis after various treatments as indicated above (Figure 5d). CT26 cells incubated with ICG-MB nanoparticles in the dark showed a high survival rate of 86.0%. With the help of 670 nm or 808 nm laser irradiation, ICG-MB obviously reduced the survival rate but induced a marked increase in the apoptosis ratio. Remarkably, ICG-MB incorporated with dual light irradiation resulted in the lowest survival percentage with the highest apoptosis level, which was in good agreement with the results from the CCK-8 assay and LIVE/DEAD cell staining. 

### 3.6. In Vivo Phototherapy Effect

Before carrying out the antitumor study, we firstly explore the biodistribution of ICG-MB to determine whether ICG-MB can effectively accumulate in the tumor site for exerting the phototherapy effect. As previously reported, ICG had a fluorescence signal extended to the NIR II region with low autofluorescence interference upon 808 nm laser irradiation [42]; we thus collected the NIR-II signal of the mice after intravenous injection of ICG-MB by using an animal NIR-II imaging system. As shown in Figure 6a, ICG-MB nanoparticles were found to spread in the whole body of the mice within 4 h. As the time increased, the fluorescence signal largely concentrated in certain areas of the mice owing to the constant circulation and metabolism. Of special note, the fluorescence signal could still be observed at the tumor site after 48 h, suggesting the preferential accumulation of ICG-MB through the EPR effect. Thereafter, the mice were sacrificed and the major organs were extracted for ex vivo imaging (Figure 6b). Although a large amount of ICG-MB nanoparticles were inevitably captured by the liver as the main metabolic organ, the tumor site also presented a distinguishable signal, suggesting the high potential of ICG-MB for NIR-II imaging guided the PDT/PTT combination therapy that followed.

The anticancer effect of ICG-MB NPs was then performed in vivo. The mice were divided into five groups: (1) PBS, (2) ICG-MB, (3) ICG-MB + 670 nm, (4) ICG-MB + 808 nm, and (5) ICG-MB + 670 nm + 808 nm. The mice were irradiated with 670 nm (20 mW/cm^2^) and 808 nm lasers (1.0 W/cm^2^) for 10 min after injecting nanoparticles for 24 h. The local heating phenomenon at the tumor region was monitored by an IR thermal camera. As exhibited in Figure 7a, the tumor site of the mice receiving systemic administration of ICG-MB was heated rapidly under 808 nm irradiation for 10 min (group 4 and group 5), demonstrating the efficient in vivo photothermal conversion behavior of ICG-MB after accumulation in tumor tissue to induce cancer cell death. Meanwhile, this local temperature elevation controlled by the external laser could effectively prevent excessive damage to healthy tissue. Furthermore, we also found that the 670 nm laser irradiation for PDT with low photodensity had limited influence on the PTT effect as the temperature of tumor site showed a similar increase trend in group 4 and group 5. The tumor size and weight of mice were observed to investigate the phototherapy effect of ICG-MB (Figure 7b–d). Compared with the PBS and ICG-MB group, PDT (group 3) or PTT (group 4) alone could significantly inhibit the growth of tumors, but the PTT effect of ICG-MB under the 808 nm laser was more robust than PDT under 670 nm. Most importantly, ICG-MB + 670 nm + 808 nm combinational therapy achieved a better curing effect than single PDT or PTT treatment. Although the tumors of group 5 have a slight increase at the initial several days due to tissue swelling after phototherapy, they almost disappeared during the following days. At the end of the experiment, mice were sacrificed and tumors were taken out to take photos. As anticipated, the size of the tumors in the combination therapy group was smaller than those in the other groups. In order to further confirm the therapy effect, we performed H&E and Ki-67 staining of tumor tissue taken from different treatment groups after laser irradiation (Figure 7e). Among all groups, ICG-MB + 670 nm + 808 nm resulted in complete loss of integral cell morphology in the H&E image and a negligible Ki67 positive signal of proliferation activity in the Ki-67 image. These results suggest that ICG-MB holds great promise for PDT and PTT combinational therapy to restrain tumor growth. Although phototherapy is mainly applied in superficial diseases located in the skin and eye due to the shallow light penetration, this technique still has application prospects in other diseases like colorectal cancer when combined with endoscopy to provide excitation light to trigger the therapeutic potential. In addition, there was no significant difference of body weight among these five groups, which meant that our treatment was safe (Figure 7f).

### 3.7. Biocompatibility Assay

The accumulations of nanoparticles in the main organs may induce acute side effects or a long-term inflammatory response in living organisms, which will hinder their biomedical applications. To explore the biocompatibility of ICG-MB, the main organs of the heart, liver, spleen, lung and kidneys were collected from the mice after different treatments for H&E staining, and the results demonstrated no obvious abnormality in these samples (Figure 8a) due to the guaranteed safety of ICG and MB in ICG-MB as FDA-approved agents, as well as the low toxicity shield from light. In addition, we carried out a blood analysis to investigate the long-term toxicity of ICG-MB in healthy mice. Both liver and kidney function markers were within the normal range (Figure 8b), indicating that no obvious hepatic or kidney disorders were caused by ICG-MB in mice. Thus, the excellent biocompatibility and low long-term toxicity associated with ICG-MB make these nanoparticles promising for future biomedical applications in clinic.

## 4. Conclusions

In summary, we rationally designed and synthesized an “all-in-one” nano-system of ICG-MB which possessed NIR-II fluorescence imaging and PTT/PDT synergistic therapeutic functions, by using the self-assembly of two FDA-approved small molecules dyes of ICG and MB. In comparison with free ICG and MB, ICG-MB was demonstrated to improve the photobleaching, photothermal conversion efficiency and ^1^O_2_ generation abilities, in addition to act as an integrated entity with an extremely high encapsulating rate and good stability in the aqueous phase. ICG-MB exhibited super cellular uptake and tumor accumulation, and in doing so carried out theranostic functions of NIR fluorescence imaging and a PTT/PDT synergistic effect in a murine colon tumor model. In addition, ICG-MB displayed no noticeable systemic toxicity as evidenced by hematological and histological analysis. Therefore, the implementation of this vehicle-free nanotheranostic provides a new strategy for the expansion and improvement of FDA-approved dyes in tumor treatment.

## Data Availability

The data presented in this study are contained within the article.

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
