# Peer review of "Vehicle-Free Nanotheranostic Self-Assembled from Clinically Approved Dyes for Cancer Fluorescence Imaging and Photothermal/Photodynamic Combinational Therapy"

_pharmaceutics, 2022, doi:10.3390/pharmaceutics14051074_

Round 1

Reviewer 1 Report

Xue et al and his co researchers designed a vehicle-free nanotheranostic self-assembled platform for cancer fluorescence imaging and photothermal/photodynamic combinational therapy. Overall, the work was well designed and demonstrated in both in vitro and in vivo level. Therefore, I recommend it for publication after solving following concerns.

  1. To prove effective conjugation of ICG and MB, it is recommended to provide FTIR evidence.
  2. is there any hydrodynamic size changes of MB observed before and after adding ICG?
  3. is the self-assembled ICG-MB stable? should run absorption spectra at different incubation points.
  4. Authors observed temp rise on ICG-MB conc. start from 10 µg/mL (Figure 2), whereas in vitro utilized (1, 2, 3, 4, 5 µg/mL) and in vivo used 4 µg/mL.
  5. Authors should check temperature elevation by using 1, 2, 3, 4, 5 µg/mL of ICG-MB.
  6. 4 µg/mL is this conc. is enough to achieve in vivo therapeutic effects, since it doesn’t have target, so it is expected that all the amount wont reach to tumor site.
  7. Why ICG-MB is more resistant to photobleaching? explain briefly?
  8. In vivo, authors utilized two lasers (670nm 20mW/cm2, 808nm 1W/cm2 for 5 minutes) to achieve therapeutic effects. Why did the authors not used same power intensity?
  9. In introduction, phototherapy section includes some recent papers (cite ; Biomater. Sci., 2021,9, 5472-5483, ACS Appl. Nano Mater. 2022, 5, 2, 1719–1733)

Author Response

Reviewer 1

Xue et al and his co researchers designed a vehicle-free nanotheranostic self-assembled platform for cancer fluorescence imaging and photothermal/photodynamic combinational therapy. Overall, the work was well designed and demonstrated in both in vitro and in vivo level. Therefore, I recommend it for publication after solving following concerns.

Response: Thanks very much for the reviewer’s positive comments. Now, we have carefully revised the manuscript according to the reviewer’s suggestions.

  1. To prove effective conjugation of ICG and MB, it is recommended to provide FTIR evidence.

Response: Thanks very much. Actually, the FT-IR spectra of free ICG, free MB and ICG-MB had been display in Figure 1f in our manuscript. The characteristic peaks at 1353 cm-1 (sulfonic group) and 1420 cm-1 (C=C stretches in long chain alkenes) in ICG, as well as the peak at 1591 cm-1 (C=N stretches in central ring) in MB, all appeared in the spectrum of ICG-MB. These results demonstrated the successful formation of hybrid nanomedicine self-assembly from ICG and MB.

  1. is there any hydrodynamic size changes of MB observed before and after adding ICG?

Response: Thanks a lot. As MB is a hydrophilic molecule which can completely dissolve in aqueous solution, so we cannot measure its hydrodynamic size, as shown in Figure Q1 below.

Figure Q1. DLS analysis of free MB in water.

  1. is the self-assembled ICG-MB stable? should run absorption spectra at different incubation points.

Response: Thanks a lot for this very important comment. According to the reviewer’s suggestion, we compared the absorption spectra of ICG-MB storied at different periods ranged from 0 to 6 day, as shown in Figure 1h. The relevant explanations were added in page 7, line 271-272. “Meanwhile, the absorption spectra of ICG-MB also did not change after storied for 6 days, further verifying the stability of this nanoagent (Figure 1f).”

  1. Authors observed temp rise on ICG-MB conc. start from 10 µg/mL (Figure 2), whereas in vitro utilized (1, 2, 3, 4, 5 µg/mL) and in vivo used 4 µg/mL.
  2. Authors should check temperature elevation by using 1, 2, 3, 4, 5 µg/mL of ICG-MB.
  3. 4 µg/mL is this conc. is enough to achieve in vivo therapeutic effects, since it doesn’t have target, so it is expected that all the amount wont reach to tumor site.

Response: I am sorry for this mistake. Indeed, the concentration of ICG-MB used in In Vitro Cytotoxicity Assay (section 2.9) and In Vitro Cell Killing Ability (section 2.10) were 10, 20, 30, 40, 50 µg/mL. In addition, the injection dose of ICG-MB in vivo evaluation was maintained at 100 μL (5 mg/mL) per mouse, as we had provided in section 2.12. We had corrected these errors in the revised manuscript.

  1. Why ICG-MB is more resistant to photobleaching? explain briefly?

Response: Thanks a lot. ICG-MB nanoparticles can reduce photobleaching of ICG small molecules via an activated quenched state possibly due to intermolecular dye interactions (Chem. Soc. Rev. 2019, 48, 2053-2108). These explanations have been added in line 299-300.

  1. In vivo, authors utilized two lasers (670nm 20mW/cm2, 808nm 1W/cm2 for 5 minutes) to achieve therapeutic effects. Why did the authors not used same power intensity?

Response: Thanks a lot for this very important comment. Indeed, the laser power intensity in our work was selected just following with other literatures about MB-based PDT and ICG-based PTT (Small 2021, 17, 2103569. J. Mater. Chem. B, 2021, 9, 9961-9970.). Especially, 20 mW/cm2 is sufficient to achieve PDT effect as demonstrated in vitro and in vivo experiment in our work. However, the density for PTT should be maintained at a relative high level from the aspect of energy conversion from light source to heat source.

  1. In introduction, phototherapy section includes some recent papers (cite ; Biomater. Sci., 2021,9, 5472-5483, ACS Appl. Nano Mater. 2022, 5, 2, 1719-1733)

Response: Thanks very much. We have carefully read these recent publications and added then as the references in the revised manuscript (ref. 8 and 9).

Reviewer 2 Report

In their paper, the authors present the results obtained with  a 
nanoplatform (named ICG-MB) obtained from two dyes: indocyanine green (ICG) and methylene blue (MB) for cancer fluorescence imaging and combinational PTT/PDT.
Authors found that the ICG-MB was found to exhibit good dispersion in the aqueous phase and improve the photostability and cellular uptake of free ICG and MB, thus exhibiting enhanced photothermal conversion and singlet oxygen generation abilities to robustly ablate cancer cells irradiated with a laser, at  808 nm and 670 nm. The product ICG-MB  was administered intravenously, and under the action of infrared radiation, has accumulated inside of tutors. Results obtained indicated that   ICG-MB shows a  combinational PTT/PDT effect, with effect in inhibition of tumor growth, without toxicities on lab animals.  The procedure presented by the authors provide a promising candidate for cancer phototherapy using currently available compounds, and for this reason, the article is interesting for scientist from oncology. However, minor corrections must be made before publishing:
1) authors must read with the attention their   article because:
-  the punctuations sign from all text of the manuscript  are not correctly inserted in the text; 
-the units of measurements are not  written correctly;
- the cell densities must be written correctly;
- the formula of conversion efficiency ( row 149-150) must be revised;
2) The sentence from the rows 211-212 must be revised;
3)Sentence from rows 213-214 must be revised (written in scientific form); the authors must write in the text in which way the lab animals were sacrificed;
4) Sentence from the row 423 must be revised
5) The sentence from the rows 450-451 must be revised
6) Conclusions must be rewritten scientifically and must reflect the main aspects of the results.

Author Response

In their paper, the authors present the results obtained with a nanoplatform (named ICG-MB) obtained from two dyes: indocyanine green (ICG) and methylene blue (MB) for cancer fluorescence imaging and combinational PTT/PDT.
    Authors found that the ICG-MB was found to exhibit good dispersion in the aqueous phase and improve the photostability and cellular uptake of free ICG and MB, thus exhibiting enhanced photothermal conversion and singlet oxygen generation abilities to robustly ablate cancer cells irradiated with a laser, at 808 nm and 670 nm. The product ICG-MB was administered intravenously, and under the action of infrared radiation, has accumulated inside of tutors. Results obtained indicated that ICG-MB shows a combinational PTT/PDT effect, with effect in inhibition of tumor growth, without toxicities on lab animals.  The procedure presented by the authors provide a promising candidate for cancer phototherapy using currently available compounds, and for this reason, the article is interesting for scientist from oncology. However, minor corrections must be made before publishing:

Response: Thanks very much for the time and efforts of the reviewer.

1) authors must read with the attention their article because:
- the punctuations sign from all text of the manuscript are not correctly inserted in the text; 
-the units of measurements are not written correctly;
- the cell densities must be written correctly;
- the formula of conversion efficiency ( row 149-150) must be revised;

Response: Thanks very much. The manuscript text has been carefully checked more thoroughly to avoid these errors.

2) The sentence from the rows 211-212 must be revised;

Response: Thanks very much. This sentence has been changed as “To explore the biodistribution in vivo, CT26 tumor-bearing mice were intravenously (i.v.) injected with ICG-MB at a dose of 100 μL (5 mg/mL) per mouse.”

3) Sentence from rows 213-214 must be revised (written in scientific form); the authors must write in the text in which way the lab animals were sacrificed;

Response: Thanks very much. This sentence has been changed as “Then an UniNano- NIR-II system was used to monitor the fluorescent signals of mice at 4, 6, 8, 10, 24, and 48 hours postinjection. After 48 h of injection, the mice were sacrificed and the main organs were resected and imaged.”

4) Sentence from the row 423 must be revised

Response: Thanks very much. This sentence has been changed as “Most importantly, ICG-MB + 670 nm + 808 nm combinational therapy…”

5) The sentence from the rows 450-451 must be revised

Response: Thanks very much. This sentence has been changed as “Both liver and kidney function markers were within the normal range (Figure. 8b), …”

6) Conclusions must be rewritten scientifically and must reflect the main aspects of the results.

Response: Thanks very much. Following the reviewer’s suggestion, the conclusions have been rewritten in the revised manuscript.

Reviewer 3 Report

In the paper entitled "Vehicle-free nanotheranostic self-assembled from clinically approved dyes for cancer fluorescence imaging and photothermal/photodynamic combinational therapy" the author describe a new nanosystem obtained from dyes with the aim of cancer imaging as well as the use of photothermal/photodynamic therapy.

The subject is of grate interest and potential in the field of cancer treatment. The article is well written. The introduction was on point and the material and methods are clearly presented. The results are reported in detail , however there are some issues that need to be addressed.

The authors have to check carefully the subscript and superscript for some of the chemical formulas and/or measurement units, etc.

Also, the references 43 to 45 are not found in the manuscript.

Finally, in the results and discussion section the results are well reported, however the data obtained are not discussed in comparison to other studies in the literature. The authors need to say how their results are positioned in comparison with the data reported previously.

Author Response

In the paper entitled "Vehicle-free nanotheranostic self-assembled from clinically approved dyes for cancer fluorescence imaging and photothermal/photodynamic combinational therapy" the author describe a new nanosystem obtained from dyes with the aim of cancer imaging as well as the use of photothermal/photodynamic therapy.

The subject is of grate interest and potential in the field of cancer treatment. The article is well written. The introduction was on point and the material and methods are clearly presented. The results are reported in detail, however there are some issues that need to be addressed.

Response: Thanks very much for the reviewer’s positive comments. Now, we have carefully revised the manuscript according to the reviewer’s suggestion.

The authors have to check carefully the subscript and superscript for some of the chemical mical formulas and/or measurement units, etc.
Response: Thanks very much. The manuscript text has been carefully checked more thoroughly to avoid these errors.
Also, the references 43 to 45 are not found in the manuscript.
Response: Thanks very much. The references 43 to 45 have been carefully checked in the revised manuscript.
Finally, in the results and discussion section the results are well reported, however the data obtained are not discussed in comparison to other studies in the literature. The authors need to say how their results are positioned in comparison with the data reported previously.
Response: Thanks very much. Actually, we had compared the photothermal conversion efficiency (η) of ICG-MB with other previously reported PTT agents, as presented in line 285-287. However, the in vitro and in vivo anticancer effect are not easy to compare because the different cancer cells and tumor models are used in different literatures. Even so, the main advantage of this work is to provide a minimalist, effective and biocompatible nanotheranostic for cancer phototherapy based on current available dyes in clinic.

Reviewer 4 Report

I have read the manuscript “Vehicle-free nanotheranostic self-assembled from clinically approved dyes for cancer fluorescence imaging and photothermal/photodynamic combinational therapy” by Mingbin Huang et al. (MS # pharmaceutics-1703159) submitted for the publication in Pharmaceutics.

In their manuscript the authors reported the preparation and characterization of a two-dyes complex suitable for PTT/PDT combined therapy. Results showed a synergistic effect against cancer in the presence of such complex.

The manuscript is interesting for the potential applications related to the absence of any additive and deserves publication in Pharmaceutics after minor revision. In particular:

  1. There are several misprints all over the manuscript (missing subscripts and upperscripts, form, nanothernostic, equations in line 149 and 187, definition of A1, ICG@MB, line 211, line 228 P should be lowercase and italics, variation, storied, x-axis in Figure 2.d and Figure 3.c, reference numbers without brackets, autofluorensce, kedney,…) ;
  2. Please, justify the use of CT26 cells as the PTT/PDT combined therapy is quite difficult to be applied in the case of colorectal cancer;
  3. Line 233; please, specify how such optimal feed ratio was found;
  4. Figure 7: How large was the increase of temperature on the tumour site during laser irradiation?
  5. What about the concentrations reported in Line 278 and the used powers for 670 and 808 nm lasers? What happens for different values?
  6. Figure 1.b and Line 237: SEM picture shows that complexes are irregular aggregates of primary spherical nanoparticles.

Author Response

I have read the manuscript “Vehicle-free nanotheranostic self-assembled from clinically approved dyes for cancer fluorescence imaging and photothermal/photodynamic combinational therapy” by Mingbin Huang et al. (MS # pharmaceutics-1703159) submitted for the publication in Pharmaceutics.

In their manuscript the authors reported the preparation and characterization of a two-dyes complex suitable for PTT/PDT combined therapy. Results showed a synergistic effect against cancer in the presence of such complex.

The manuscript is interesting for the potential applications related to the absence of any additive and deserves publication in Pharmaceutics after minor revision. In particular:

  1. There are several misprints all over the manuscript (missing subscripts and upperscripts, form, nanothernostic, equations in line 149 and 187, definition of A1, ICG@MB, line 211, line 228 P should be lowercase and italics, variation, storied, x-axis in Figure 2.d and Figure 3.c, reference numbers without brackets, autofluorensce, kedney,…) ;

Response: Thanks very much. The manuscript text has been carefully checked more thoroughly to avoid these errors.

  1. Please, justify the use of CT26 cells as the PTT/PDT combined therapy is quite difficult to be applied in the case of colorectal cancer;

Response: Thanks very much. According to the reviewer’s suggestion, we have added some more explantions about the future prospect of PTT/PDT in colorectal cancer tumor in page 13, line 453-456 as follows: “Although phototherapy is mainly applied in superficial diseases located in skin and eye due to the shallow light penetration, this technic still has application prospects in other diseases like colorectal cancer when combined with endoscopy to provide excitation light to trigger therapeutic potential.”

  1. Line 233; please, specify how such optimal feed ratio was found;

Response: Thanks very much. According to the reviewer’s suggestion, we have added more words to explain how we selected optimal feed ration in page 6, line 239-241 in the revised manuscript as follows: “In this work, ICG-MB nanosystems were synthesized via a facile nano-precipitation method at the optimized feed ratio of 2:1 (ICG:MB, W/W) to provide a stable nanoparticulated dispersion with distinct Tyndall phenomenon and without aggregation after storied above 24 h.”

  1. Figure 7: How large was the increase of temperature on the tumour site during laser irradiation?

Response: Thanks very much. Following the reviewer’s suggestion, the temperature on the tumour site during laser irradiation have been labeled in the Figure. 7a in the revised manuscript.

  1. What about the concentrations reported in Line 278 and the used powers for 670 and 808 nm lasers? What happens for different values?

Response: Thanks very much. Indeed, the laser power intensity in our work was selected just following with other literatures about MB-based PDT and ICG-based PTT. Especially, 20 mW/cm2 is sufficient to achieve PDT effect as demonstrated in vitro and in vivo experiment in our work. However, the density for PTT should be maintained at a relative high level from the aspect of energy conversion from light source to heat source.

  1. Figure 1.b and Line 237: SEM picture shows that complexes are irregular aggregates of primary spherical nanoparticles.

Response: Thanks very much. We have revised the presentations in the revised manuscript.

Round 2

Reviewer 1 Report

Authors resolved all my preveious concerns and now I recommend it for publication in Pharmaceutics journal.

Reviewer 3 Report

The manuscript improved and the paper can be accepted in the present form.